# Tuberculosis-Related Hospitalizations in a Low-Incidence Country: A Retrospective Analysis in Two Italian Infectious Diseases Wards

**DOI:** 10.3390/ijerph17010124

**Published:** 2019-12-23

**Authors:** Laura Campogiani, Mirko Compagno, Luigi Coppola, Vincenzo Malagnino, Gaetano Maffongelli, Lavinia Maria Saraca, Daniela Francisci, Franco Baldelli, Carla Fontana, Sandro Grelli, Massimo Andreoni, Giovanni Sotgiu, Laura Saderi, Loredana Sarmati

**Affiliations:** 1Clinical Infectious Diseases, Department of System Medicine, Tor Vergata University, 00133 Rome, Italy; lauracampg@gmail.com (L.C.); mirkocompagno2@gmail.com (M.C.); luigi.coppolamed@gmail.com (L.C.); malagninovincenzo@gmail.com (V.M.); andreoni@uniroma2.it (M.A.); 2Clinical Infectious Diseases, Belcolle Hospital, 01100 Viterbo, Italy; maffongelligaetano@gmail.com; 3Clinic of Infectious Diseases, Department of Medicine, University of Perugia, 06123 Perugia, Italy; lavitrinity@libero.it (L.M.S.); Daniela.francisci@unipg.it (D.F.); franco.baldelli@unipg.it (F.B.); 4Laboratory of Microbiology, Policlinico Tor Vergata, 00133 Rome, Italy; carla.fontana@uniroma2.it; 5Department of Experimental Medicine, Tor Vergata University, 00133 Rome, Italy; grelli@med.uniroma2.it; 6Clinical Epidemiology and Medical Statistics Unit, Department of Medical, Surgical and Experimental Sciences, University of Sassari, 07100 Sassari, Italy; gsotgiu@uniss.it (G.S.); lsaderi@uniss.it (L.S.)

**Keywords:** tuberculosis, pulmonary tuberculosis, extrapulmonary tuberculosis, foreign people

## Abstract

In recent years, a decrease in the incidence of tuberculosis (TB) has been recorded worldwide. However, an increase in TB cases has been reported in foreign people living in low-incidence countries, with an increase in extrapulmonary TB (EPTB) in the western region of the world. In the present work, a retrospective study was conducted in two Italian infectious diseases wards to evaluate the clinical characteristics of TB admission in the time period 2013–2017. A significant increase in TB was shown in the study period: 166 (71% males) patients with TB were enrolled, with ~70% coming from outside Italy (30% from Africa, 25% from Europe, and 13% from Asia and South America). Compared to foreign people, Italians were significantly older (71.5 (interquartile range, IQR: 44.5–80.0) vs. 30 (IQR: 24–40) years; *p* < 0.0001) more immunocompromised (48% vs. 17%; *p* < 0.0001), and affected by comorbidities (44% vs. 14%; *p* < 0.0001). EPTB represented 37% of all forms of the disease, and it was more incident in subjects coming from Africa than in those coming from Europe (39.3% vs. 20%, respectively). In logistic regression analysis, being European was protective (odd ratio, OR (95% CI): 0.2 (0.1–0.6); *p* = 0.004) against the development of EPTB forms. In conclusion, an increase in the rate of TB diagnosis was documented in two Italian reference centers in the period 2013–2017, with 39% of EPTB diagnosed in patients from outside Europe.

## 1. Introduction

Tuberculosis (TB) is an important clinical and public health issue worldwide. The majority (87%) of new TB cases in 2017 occurred in 30 high TB-burden countries (20 countries with the highest absolute number of TB and 10 countries defined by >10,000 estimated incident TB cases/year), whereas only 6% of cases were estimated in the World Health Organization (WHO) European and American Regions [1].

During the last decade, a decreased TB notification rate has been recorded in the USA and in Western European countries, making the WHO target of TB elimination potentially achievable in those geographical areas. However, two different trends have been detected in countries with low TB incidence: decreased and increased TB incidence trends in autochthonous and foreign-born populations, respectively [2,3]. This has also been proven in American children and adolescents, where race and ethnicity disparity in the TB rates has been shown [4]. The higher TB incidence in foreign-born persons living in low-incidence countries is associated with the high TB incidence in their country of origin, more often following reactivation of a latent TB infection owing to poor living conditions in the host country. Moreover, the increased risk of *Mycobacterium tuberculosis* infection can be associated with travels to the native country, involving new generations of foreign-born families [5]. Cowger and colleagues showed that having at least one parent born outside of the USA increased the likelihood of developing TB by 3.5 times, while having two parents born outside of the USA increased the risk up to 8.5 times [4].

Italy is a country with low TB incidence (i.e., <10 per 100,000 population annually), with 4100 new cases estimated in 2017 [1]. However, an increased migratory flow, mainly from countries with high TB incidence, has been documented since 2015 (i.e., from 50,000 migrants in 2013 to >150,000 in 2014–2015 and ~200,000 in 2016) [6]. From 2004 to 2014, >50% of the annual Italian TB notifications involved foreigners, which was correlated to the increased number of migrants [7]. Few data are available on the anatomical localizations of TB disease in the migrant population in Italy, but it is known that migrants, together with immunocompromised and human immunodeficiency virus (HIV)-positive patients, are at higher risk of extrapulmonary TB (EPTB). The diagnosis of EPTB can be difficult, and its treatment duration is long to avoid any recurrences. EPTB is also reported to occur more frequently in younger subjects, particularly in children, where the difficulty of diagnosis and a correct therapeutic approach is often the cause of serious clinical sequelae. Recently, a survey conducted in the USA showed that EPTB is becoming more incident, although TB hospitalization rates are declining [8].

The aim of the present study is to describe the trend, as well as the epidemiological and clinical characteristics, of TB-related hospital admissions in two university hospitals in Italy from 2013 to 2017 and to assess the clinical correlation of pulmonary TB (PTB) and EPTB.

## 2. Methods

A retrospective cohort study was performed at Tor Vergata Hospital, Rome, and Perugia University Hospital, Perugia, Italy. Patients with a TB diagnosis admitted to infectious disease wards between January 2013 and December 2017 were recruited. The study included all hospitalized patients with TB. Patients were Italians, legal foreign residents, and undocumented immigrants. Undocumented immigrants with TB have access to urgent and essential hospital services through emergency rooms. All enrolled patients were admitted in the context of the free assistance offered by the Italian National Health Service.

An ad hoc electronic database was created to collect demographic (i.e., sex, age, and country of origin), epidemiological (i.e., comorbidities (copresence of ischemic heart disease, atherosclerotic vasculopathy, hypertension, and chronic pulmonary disease) and alcohol and/or drug abuse), and clinical (i.e., symptoms and clinical signs, radiological signs, microbiological results, and therapy) variables from medical files.

TB diagnosis was based on the standards of the European Center for Disease Prevention and Control (ECDC) and the European Respiratory Society (ERS) [9,10]. Patients were initially screened for *Mycobacterium tuberculosis* infection with the tuberculin skin test, the interferon-gamma release assay (IGRA) (QuantiFERON^®^-TB Gold Plus, Diasorin, Vicenza, Italy), or both tests. All patients with suspected PTB underwent sputum testing either by microscopy using Zhiel Neelsen staining or the Xpert^®^ MTB/RIF Ultra assay (Cepheid, Sunnyvale, CA, USA). For the other specimens and to detect the resistance genes (first- and second-line drugs) in *Mycobacterium tuberculosis*-positive specimens, the Anyplex™ MTB/NTM and Anyplex™ MTB/NTMe Real-Time Detection and Anyplex™ II MTB/MDR/XDR Detection (Seegene, Rockville, Maryland, USA) were used, following the manufacturer’s instructions (evaluating susceptibility to rifampicin, isoniazid, fluoroquinolones, and aminoglycosides). Positive specimens (such as sputum, bronchoalveolar lavage, pleural fluid, cerebrospinal fluid, peritoneal fluid, urine, blood, and purulent material from abscesses) and/or tissue biopsies were digested and decontaminated prior to culture according to standard protocols [11]. Mycobacterial culture was performed on a solid (Lowenstein–Jensen tubes, bioMérieux, Cambridge, MA, USA) and a liquid medium (automated Bactec MGIT system, Becton Dickinson, Salt Lake City, Utah, USA). Phenotypic drug susceptibility testing (DST) to rifampicin, isoniazid, pyrazinamide, ethambutol, and streptomycin was performed using the MGIT 960 platform in conjunction with EpiCenter software equipped with the TB eXiST module (Becton Dickinson) according to the manufacturer’s recommendations.

All personal information was treated confidentially and anonymously. The Tor Vergata Ethics Committee approved the study.

### Statistical Analysis

Qualitative and quantitative variables were summarized with absolute and relative (percentages) frequencies and means (standard deviations, SD) or medians (interquartile ranges, IQR) depending on their parametric distribution. In-between group differences were statistically assessed with the chi-squared or Fisher’s exact test, when appropriate, for qualitative variables, while the Student’s *t* test and the Mann–Whitney test were used for parametric and nonparametric quantitative variables, respectively. Logistic regression analyses were carried out to assess the relationship between outcome variables and demographic, epidemiological, and clinical variables. A two-tailed *p*-value was considered statistically significant if less than 0.05. The statistical software STATA version 16 (StatsCorp, College Station, TX, USA) was used to perform all statistical analyses.

## 3. Results

A significant increase in TB diagnosis in the two Italian hospitals during the period 2013–2016 was found (Table 1 and Appendix A). Overall, 166 patients were enrolled; 71% (*n* = 118) were male, with a median (IQR) age of 37 (26–55) years (Table 1). Only one-third (31.3%) were Italians, whereas ~70% came from African (30%), European (25%, mainly Eastern European countries), and other countries of the world (13%, mainly Asian and South American countries). From 2013 to 2017, there was an increased number of TB notifications, with a slight decline in 2017 compared to 2016. Forty-five (27%) patients were immunocompromised, one-third of whom (31%) were HIV-positive. The majority of patients complained of fever (62.4%), cough (54%), and weight loss (52%); only 12.7% had hemoptysis (Appendix A). Most of them showed tuberculin skin testing (91%) and/or IGRA (89%) positivity (Appendix A). The percentages of PTB and EPTB were 57% and 37%, respectively, whereas the remaining 6% were mixed forms (Table 1). The prevalent chest radiological finding was lobar consolidation (56.5%), and only a minority (19.3%) showed the tree-in-bud radiological picture (Appendix A). Drug resistance data were retrievable for 126 patients. In three cases, both molecular and phenotypic test were done (Appendix A). Sixteen (13%) of the 126 patients had drug-resistant TB, with mycobacterial strains mainly resistant to pyrazinamide, isoniazid, and rifampicin (Appendix A). The majority (14; 87%) of patients with drug-resistant TB were foreigners. Four (3.3%) subjects (three from Africa and one from Eastern Europe) had forms caused by *M. tuberculosis* strains resistant to more than one TB drug.

Foreign-born patients were significantly younger (median (IQR) age: 30 (24–40) years versus 71.5 (44.5–80.0) years; *p* < 0.0001), mainly males, and more frequently affected by drug-resistant TB forms (17.1% versus 2.9%; *p* = 0.04) (Table 1). Conversely, Italians were significantly more immunocompromised (48% versus 17%; *p* < 0.0001) or affected by more comorbidities (44% versus 14%; *p* < 0.0001). However, a higher percentage of HIV-positive patients with TB were foreign-born (10 (50%) versus 4 (16%); *p* = 0.006).

The temporal distribution of TB hospitalizations and the proportion of PTB, EPTB, and the PTB + EPTB mixed forms are shown in Figure 1. Excluding mixed forms (10 cases PTB + EPTB), almost 40% of the study sample had EPTB, and of the 61 patients with EPTB, 43 (70%) were foreign-born (Table 1). A higher annual prevalence of EPTB in foreigners was shown during the study period, with the lowest and highest values in 2015 (62%) and 2017 (91%), respectively (Figure 2A). The most prevalent EPTB anatomical localization was lymph nodes, followed by osteoarticular, gastrointestinal, and pleural sites (Figure 2B). Five EPTB cases had more than one anatomical site of localization (described in the table on the bottom of Figure 2B).

EPTB was significantly more incident in foreign patients coming from Africa than in those coming from Europe (24 (39.3%) and 7 (11.5%); *p* = 0.01 and 0.001, respectively) (Table 2). EPTB was significantly more frequent in subjects with comorbidities (21 (34.4%); *p* = 0.01) and with immunodepression (23 (37.7%); *p* = 0.006).

In the logistic regression analysis (Table 3), being European seemed to be protective (odds ratio, OR (95% confidence interval, CI): 0.2 (0.1–0.6); *p* = 0.004) against the occurrence of EPTB forms.

## 4. Discussion

Our results showed an increased rate of TB diagnosis in two reference Italian hospitals during the time period 2013–2017. Overall, the autochthonous population accounted for only one-third of all hospitalized TB cases, and they were more frequently older (>70 years old) and affected by immunodepression or other comorbidities. Almost 40% of the study subjects showed EPTB, which was more incident in young males of non-European origin. Approximately one third of the EPTB patients were immunocompromised (mainly HIV-positive or affected by hematological malignancy).

A decreased TB incidence has been proven in all WHO regions, with more relevant findings in countries with high TB incidence [1]. WHO/Europe showed a 4.5% annual decrease in the TB notification rate from 2013 to 2017 [12]. The decline in the national incidence rate was attenuated by the increased TB incidence in the foreign-born population living in European countries (from 27.1% in 2013 to 33.1% in 2017) [2,5,12,13,14,15,16,17,18,19].

The increased TB-related hospitalization rate, shown in the two reference Italian clinical centers in the same period of 2013–2017, was mainly attributed to the increased arrival of asylum seekers and political/economic migrants who escaped from geographical areas with conflicts and governmental instability [20]. The demographic and epidemiological characteristics of foreign-born patients with TB in our study are similar to those described by other European studies [13,14,15,16,17,18,19]. Economic and financial constraints could have driven the migration of the youngest people, especially males [21]; stress conditions, including the long travel from African and Asian countries and the poor living conditions on arrival, might have favored *Mycobacterium tuberculosis* reactivation and then the occurrence of the disease.

More than 80% of patients with drug-resistant TB and all patients with more than one drug resistance were non-Italian in our study. The correlation between drug-resistant TB and immigration from countries with high TB burden is well known, even though conditions like intravenous drug use and comorbidities as well as HIV infection have been reported as factors associated with the possible emergence of drug-resistant TB in low-incidence countries such as Italy and Spain [22].

It is noteworthy that a higher proportion of hospitalizations was due to EPTB (39%), which is in contrast to data described by other Italian studies [23]. The ECDC reported [24] that EPTB accounted for 14.2% of all TB cases in European countries in 2016, with the highest proportions (~30%) in Belgium, Ireland, the Netherlands, Norway, Sweden, Turkey, the United Kingdom, and Italy. Moreover, an increased EPTB hospitalization rate (22.3%) was recently described in the USA [8].

The country of origin and EPTB localization could have a relationship [25]. Our EPTB cases were more frequently young, male, with few comorbidities, and of African origin. A WHO/ECDC paper, which reviewed the characteristics of European EPTB cases from 2003 to 2014 [25], showed similar results, with a more frequent EPTB diagnosis in young females coming from non-European countries. The association of EPTB with the female sex is a known finding that has been confirmed by recently published African studies [26,27,28], and its origin can be found in genetic diversities and sex differences in smoking and social exposures. However, an Australian retrospective study [29] found a higher EPTB rate in males; therefore, the correlation between EPTB and sex seems to be influenced by the study population’s sociodemographic characteristics.

Only 11% of non-European EPTB patients were HIV-positive in our study; thus, other genetic, epidemiological, and clinical factors could be associated with the high EPTB rate.

Mycobacterial factors could play a role based on the heterogeneous geographic distribution of *Mycobacterium tuberculosis* strains and the higher EPTB incidence in some geographic areas. However, only a few studies have investigated this association [30,31], and currently, no specific *M. tuberculosis* strain lineage has been linked to EPTB.

Our study has several limitations. Its retrospective nature and the recruitment of only two clinical centers can increase the risk of confounding and reduce the generalizability of the findings. However, based on the low TB incidence rate in Italy and the high density of TB cases in metropolitan areas, our results can show hypothesis-generating findings, which need to be confirmed in further multicenter studies. Furthermore, the lack of data on the genetic background of migrants and on the genotype of *Mycobacterium tuberculosis* strains limits the assessment of key host- and microorganism-related variables.

## 5. Conclusions

In conclusion, an increased TB notification rate was found in two Italian reference centers in the time period 2013–2017, with 39% of EPTB diagnosed in patients coming from non-European countries. Based on the challenges of EPTB diagnosis, its protean clinical features, and the increased migration flows from countries with high TB incidence, screening and diagnostic efforts should be improved when managing foreign-born patients. Neglecting TB diagnosis and prevention in high-risk populations can hinder the epidemiological advances achieved in high-income countries.

## Figures and Tables

**Figure 1 ijerph-17-00124-f001:**
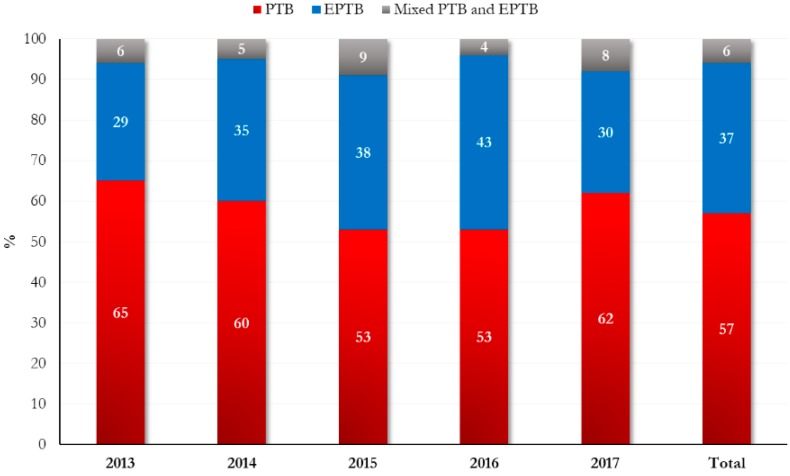
Temporal distribution of tuberculosis-related admissions from 2013 to 2017 in the study cohort.

**Figure 2 ijerph-17-00124-f002:**
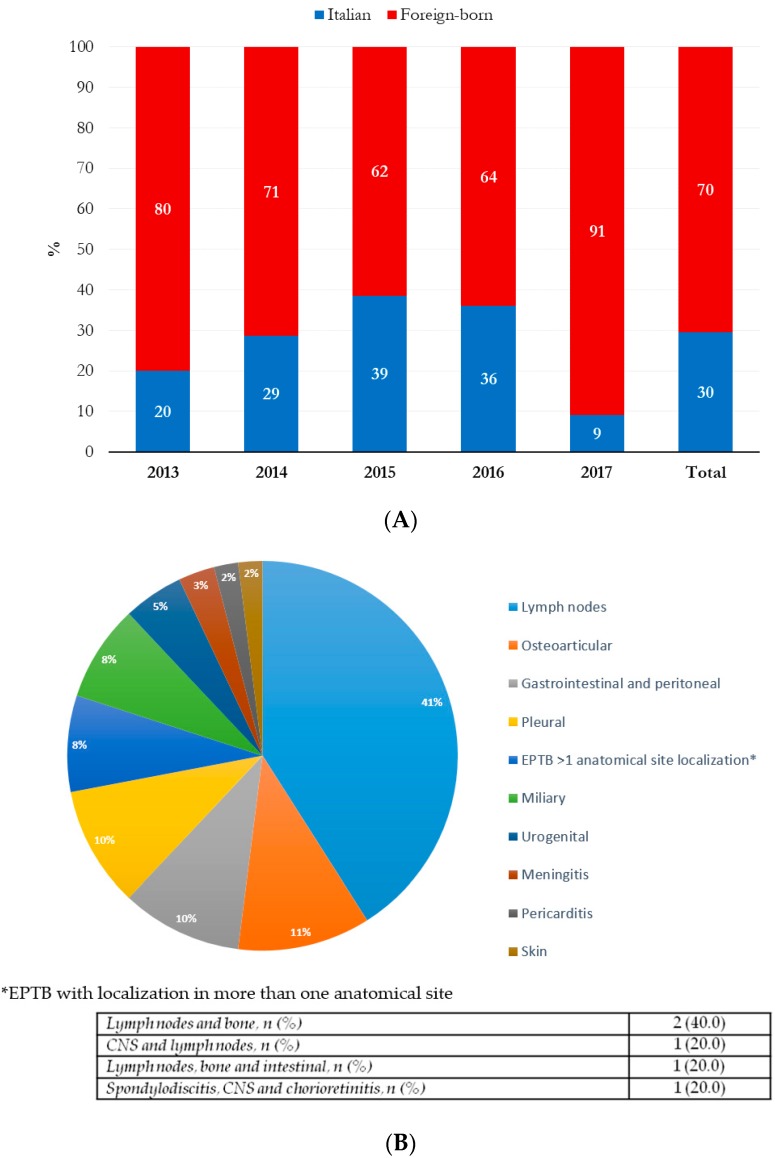
(**A**) Prevalence of EPTB between Italian and foreign-born patients in the study period 2013–2017. (**B**) Anatomical sites of EPTB forms; in the table, EPTB forms with more than one anatomical site of localization are reported.

**Table 1 ijerph-17-00124-t001:** Descriptive analysis of patients’ characteristics by their country of origin.

Variables		Overall *n* = 166	Italian (*n* = 52)	Foreign-Born (*n* = 114)	*p*-Value ^&^
Year of diagnosis, *n* (%)	2013	17 (10.2)	3 (5.8)	14 (12.3)	0.29
	2014	20 (12.1)	6 (11.5)	14 (12.3)	
	2015	34 (20.5)	13 (25.0)	21 (18.4)	
	2016	58 (34.9)	22 (42.3)	36 (31.6)	
	2017	37 (22.3)	8 (15.4)	29 (25.4)	
Median (IQR) age, years		37 (26–55)	71.5 (44.5–80.0)	30 (24–40)	<0.0001
Age >65 years		31 (18.7)	31 (59.62)	0 (0.0)	<0.0001
Age group, *n* (%)	0–24	31 (18.7)	2 (3.9)	29 (25.4)	0.001
	25–44	74 (44.6)	11 (21.2)	63 (55.3)	<0.0001
	45–64	30 (18.1)	8 (15.4)	22 (19.3)	0.55
	65–79	17 (10.2)	17 (32.7)	0 (0.0)	<0.0001
	≥80	14 (8.4)	14 (26.9)	0 (0.0)	<0.0001
Males, *n* (%)		118 (71.1)	32 (61.5)	86 (75.4)	0.07
Immunodepression, *n* (%)		45 (27.1)	25 (48.1)	20 (17.5)	<0.0001
Causes of immunodepression, *n* (%)	HIV positivity	14 (31.1)	4 (16.0)	10 (50.0)	0.01
	Hematological diseases	6 (13.3)	6 (24.0)	0 (0.0)	0.02
	Alcohol	4 (8.9)	1 (4.0)	3 (15.0)	0.20
	Diabetes mellitus	4 (8.9)	2 (8.0)	2 (10.0)	0.82
	Solid tumor	4 (8.9)	4 (16.0)	0 (0.0)	0.06
	Malnutrition	4 (8.9)	1 (4.0)	3 (15.0)	0.20
	Autoimmune disease	3 (6.7)	3 (12.0)	0 (0.0)	0.11
	Chronic renal failure	2 (4.4)	1 (4.0)	1 (5.0)	0.87
	Other diseases	4 (8.9)	3 (12.0)	1 (5.0)	0.41
Comorbidity *, *n* (%)		39 (23.5)	23 (44.2)	16 (14.0)	<0.0001
TB form, *n* (%)	PTB **	95 (57.2)	32 (61.5)	63 (55.3)	0.66
	EPTB ***	61 (36.8)	18 (34.6)	43 (37.7)	
	PTB and EPTB	10 (6.0)	2 (3.9)	8 (7.0)	
TB drug resistance, *n* (%)		16 (13.0)	1 (2.9)	15 (17.1)	0.04

* Comorbidity was defined as the copresence of ischemic heart disease, atherosclerotic vasculopathy, hypertension, or chronic pulmonary disease; ** PTB = pulmonary tuberculosis; *** EPTB = extrapulmonary tuberculosis; ^&^ between Italian and foreign-born.

**Table 2 ijerph-17-00124-t002:** Descriptive analysis of patients’ characteristics by pulmonary and extrapulmonary TB forms.

Variables		Pulmonary (*n* = 95)	Extrapulmonary (*n* = 61)	*p*-Value
Year of diagnosis, *n* (%)	2013	11 (11.6)	5 (8.2)	0.74
	2014	12 (12.6)	7 (11.5)	
	2015	18 (19.0)	13 (21.3)	
	2016	31 (32.6)	25 (41.0)	
	2017	23 (24.2)	11 (18.0)	
Median (IQR) age, years		39 (29–58)	32 (24–55)	0.07
Age >65 years		19 (20.0)	12 (19.7)	0.96
Age group, *n* (%)	0–24	10 (10.5)	16 (26.2)	0.13
	25–44	48 (50.5)	23 (37.7)	
	45–64	18 (19.0)	10 (16.4)	
	65–79	10 (10.5)	7 (11.5)	
	≥80	9 (9.5)	5 (8.2)	
Males, *n* (%)		65 (68.4)	45 (73.8)	0.48
Geographical area of origin, *n* (%)	Italy	32 (33.7)	18 (29.5)	0.59
	Africa	19 (20.0)	24 (39.3)	0.01
	Europe	35 (36.8)	7 (11.5)	0.001
	Other countries *	9 (9.5)	12 (19.7)	0.07
Immunodepression, *n* (%)		17 (17.9)	23 (37.7)	0.006
Causes of immunodepression, *n* (%)	HIV positivity	3 (17.7)	10 (43.5)	0.18
	Hematological diseases	1 (5.9)	5 (21.7)	
	Alcohol	4 (23.5)	0 (0.0)	
	Diabetes mellitus	2 (11.8)	2 (8.7)	
	Solid tumor	2 (11.8)	2 (8.7)	
	Malnutrition	0 (0.0)	1 (4.4)	
	Autoimmune disease	2 (11.8)	1 (4.4)	
	Chronic renal failure	1 (5.9)	1 (4.4)	
	Other diseases	2 (11.8)	1 (4.4)	
Comorbidity ^&^, *n* (%)		16 (16.8)	21 (34.4)	0.01

* Asian and South American countries; ^&^ comorbidity was defined as the copresence of ischemic heart disease, atherosclerotic vasculopathy, hypertension, or chronic pulmonary disease.

**Table 3 ijerph-17-00124-t003:** Logistic regression analysis to assess the relationship between demographic and clinical characteristics and extrapulmonary disease.

		Univariate Analysis		Multivariate Analysis	
		OR (95% CI)	*p*-Value	OR (95% CI)	*p*-Value
Age, years		1.0 (1.0–1.0)	0.21	1.0 (1.0–1.0)	0.98
Age >65 years		1.0 (0.4–2.2)	0.96	-	-
Age groups		0.9 (0.7–1.1)	0.29	-	-
Males		1.3 (0.6–2.7)	0.48	1.3 (0.6–2.9)	0.58
Foreign-born		1.2 (0.6–2.4)	0.59	-	-
Geographical area of origin	Italy	0.8 (0.4–1.7)	0.59	-	-
	Africa	2.6 (1.3–5.3)	0.009	0.8 (0.3–2.6)	0.76
	Eastern Europe	0.2 (0.1–0.5)	0.001	-	-
	Other countries *	2.3 (0.9–5.9)	0.07	-	-
European origin		0.3 (0.2–0.7)	<0.0001	0.2 (0.1–0.6)	0.004
HIV positivity		6.0 (1.6–22.8)	0.008	2.3 (0.4–12.1)	0.33
Hematological diseases		8.4 (1.0–73.7)	0.06	-	-
Other causes of immunodepression		1.0 (0.4–2.5)	0.91	-	-
Comorbidity ^&^		2.5 (1.2–5.3)	0.01	3.4 (1.1–10.4)	0.03

* Asian and South American countries; ^&^ comorbidity was defined as the copresence of ischemic heart disease, atherosclerotic vasculopathy, hypertension, or chronic pulmonary disease.

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
