# Peer review of "Tuberculosis-Related Hospitalizations in a Low-Incidence Country: A Retrospective Analysis in Two Italian Infectious Diseases Wards"

_ijerph, 2019, doi:10.3390/ijerph17010124_

Round 1
Reviewer 1 Report
Dr. Campogiani et al. retrospectively analyzed the hospitalized patients with clinically confirmed diagnosis of tuberculosis in two reference university hospitals in Italy during the period of 2013-2017. A total of 166 patients were enrolled for the study. They concluded that extra-pulmonary tuberculosis was more prevalent among foreign-born patients. The topic is potentially interesting because tuberculosis among foreigners are more and more important in the European region. The extrapulmonary tuberculosis is more difficult to be diagnosed and if high percentage of foreign-born patients of tuberculosis are free from pulmonary presentations, it will have great impact on the screening test and case-finding policies dealing with migrants.
Major points:
The age of distribution for tuberculosis is not normal distribution for TB patients. According to the Global Report 2019 from WHO, the age distribution of TB cases in Italy is U-shaped that two peaks of age groups (i.e. age less than 35 years, and age greater than 65 years) were noted. The observation was also found in the present report that foreign patients of TB has a median age of 30 while the Italian patients of TB have a median age of 71.5. The influence of age on the extrapulmonary tuberculosis is also non-linear. In general, extrapulmonary TB affects persons with diabetes and HIV, as well as young children (<15 years of age) and older adults (>65 years of age) (Yang, Clin Infect Dis. 2004;38:199–205). It is suggested to categorize the patients into age groups instead of using it as continuous variable. Because the two hospitals involved were both tertiary referral centers. The percentage of extra-pulmonary tuberculosis could be distorted by the health-insurance payment policy and the National TB program. The authors should briefly describe the difference among referral process of Italians and foreign patients suspected of having TB. The report enrolled only hospitalized patients, the authors should also briefly describe the policy of hospilalization for patients suspected of having TB. Are all patients hospitalized? In the supplement file, there are two p-values that I cannot figure out the meanings of these p-values. They were referred from the text in the first paragraph of the result section: "A significant increase in TB diagnosis in the two Italian hospitals during the period 2013-2017 was found (figure 1)." Please clarify the statistics used here. The European foreigners has high percentage of pulmonary manifestation as the Italian patients. Comparing the Italian vs non-Italian and the European vs non-European both showed that foreigners from non-European regions have higher percentage of extra-pulmonary presentations. The authors better focus on only one stratification scheme (either Italian vs non-Italian, or European vs non-European) to prevent distracting the audience. In the Table 5, the effect of pooled immunocompromised status was not significant in multivariate analysis. I believe that if the authors use only HIV and haematological malignancy that strongly influence the T-cell immunity as the immunocompromised status in to the statistics, these factors should be significant. In the text, only 6% of all patients were categoriezed as mixed pulmonary and extrapulmonary form. However, in the figure 1, the mixed form seemed to be more than 10% every year.Minor points
The figure 1 is suggested to be presented as percentage stacking bar chart. Using a bar chart will mislead the readers. Figure 2(A) is suggested to be presented without 3D effect. According to radiological presentations in the table 1. It seemed only 159 (of the 166 patients reported) have the result of chest radiography (91 / 0.572 = 159). For the 7 patients without chest radiography, how could the authors be sure that they are free from pulmonary tuberculosis? The authors described a 5.7% of isoniazid resistance, I am interested about the method used to determine the isoniazid drug-susceptibility. The methods described by the authors included genotypic methods (Xpert MTB/RIF, and anyPlex) and phenotypic methods. It is strange that the isoniazid, rifampicin, ethambutol, pyrazinamide and aminoglycosides have the same denominators of 122. If the specimen did not underwent DST for the specific agents, it is not appropriate to include this specimen in the denominator. There is a small table under the Figure 2(B) describing mixed extra-pulmonary anatomical distribution of 5 patients. The table is not referred by the text.Author Response
Manuscript ID: ijerph-656990
Type of manuscript: Article
Title: Tuberculosis-related hospitalizations in a low-incidence country: a retrospective analysis in two Italian infectious diseases wards
Reviewer 1
The age of distribution for tuberculosis is not normal distribution for TB patients. According to the Global Report 2019 from WHO, the age distribution of TB cases in Italy is U-shaped that two peaks of age groups (i.e. age less than 35 years, and age greater than 65 years) were noted. The observation was also found in the present report that foreign patients of TB has a median age of 30 while the Italian patients of TB have a median age of 71.5. The influence of age on the extrapulmonary tuberculosis is also non-linear. In general, extrapulmonary TB affects persons with diabetes and HIV, as well as young children (<15 years of age) and older adults (>65 years of age) (Yang, Clin Infect Dis. 2004;38:199–205). It is suggested to categorize the patients into age groups instead of using it as continuous variable.
Answer- We thank the Reviewer for the suggestion. We included the new categorization of the variable age, highlighting in the tables both the proportion of individuals aged >65 years and the age categories described in the paper cited by the Reviewer. Statistical comparisons were carried out to assess differences related to the age groups requested by the Reviewer.
Because the two hospitals involved were both tertiary referral centers. The percentage of extra-pulmonary tuberculosis could be distorted by the health-insurance payment policy and the National TB program. The authors should briefly describe the difference among referral process of Italians and foreign patients suspected of having TB. The report enrolled only hospitalized patients, the authors should also briefly describe the policy of hospitalization for patients suspected of having TB. Are all patients hospitalized?
Answer- We thank the Reviewer for the request of clarification. The Italian National Health Service covers all citizens and legal foreign residents. Coverage is automatic and universal. Since 1998, undocumented immigrants have access to urgent and essential services. Since the National Health Service does not allow people to opt out of the system and seek only private care, substitutive insurance does not exist, but complementary and supplementary private health insurance are available.
Free outpatient services exist for foreigners without regular documents, and screening programs are provided in the migrant reception network. Whenever TB is suspected, patients are sent to the Hospital by territorial network. In hospital, foreigners receive care, regardless of documentation status. Upon TB suspicion or diagnosis, subjects are referred to the Infectious Diseases ward. All patients enrolled in the study were hospitalized TB cases.
A phrase regarding the characteristics of the study populations and the health cover is now present in the methods section of the article (lines 72-75).
In the supplement file, there are two p-values that I cannot figure out the meanings of these p-values. They were referred from the text in the first paragraph of the result section: "A significant increase in TB diagnosis in the two Italian hospitals during the period 2013-2017 was found (figure 1)." Please clarify the statistics used here.
Answer- We thank the Reviewer for this clarification. We included a footnote in the Figure 1 to better clarify this issue. The p-value 0.85 refers to the comparison between the proportions of pulmonary and extra-pulmonary TB during the study period, whereas the p-value <0.0001 refers to the statistical assessment of the trend.
The European foreigners has high percentage of pulmonary manifestation as the Italian patients. Comparing the Italian vs non-Italian and the European vs non-European both showed that foreigners from non-European regions have higher percentage of extra-pulmonary presentations. The authors better focus on only one stratification scheme (either Italian vs non-Italian, or European vs non-European) to prevent distracting the audience.
Answer- We thank the Reviewer for having raised this issue. We excluded the table 4 (and the text related to, lines 158-162) where it is described the more generic comparison between Europeans and non-Europeans, to avoid any misunderstandings.
In the Table 5, the effect of pooled immunocompromised status was not significant in multivariate analysis. I believe that if the authors use only HIV and haematological malignancy that strongly influence the T-cell immunity as the immunocompromised status in to the statistics, these factors should be significant
Answer- We thank the Reviewer for this suggestion. We carried out the analysis and we did not find any role played by the variable haematological disease. So we decided to include the only variable HIV-positivity.
In the text, only 6% of all patients were categoriezed as mixed pulmonary and extrapulmonary form. However, in the figure 1, the mixed form seemed to be more than 10% every year.
Answer- Thanks for pointing this out. In the figure 1, we reported only pulmonary and extrapulmonary TB. The line reported as “combined” refers to both PTB and EPTB summed together (the % refers to TB-related hospital admission on total hospital admission, for every cause), not to the mixed forms. Mixed forms are excluded in this figure, as stated in the figure legend.
Minor points
1.The figure 1 is suggested to be presented as percentage stacking bar chart. Using a bar chart will mislead the readers. Figure 2(A) is suggested to be presented without 3D effect
Answer- We thanks the Reviewer for the suggestion. The new version of the graphs is included in the manuscript.
According to radiological presentations in the table 1. It seemed only 159 (of the 166 patients reported) have the result of chest radiography (91 / 0.572 = 159). For the 7 patients without chest radiography, how could the authors be sure that they are free from pulmonary tuberculosis?
Answer- Thanks for pointing this out. Chest radiology data were present for all but 5 patients. Some patient had more than one radiology finding simultaneously, separately evaluated in the table, others did not have chest alterations, presenting with exclusive EPTB.
Of 5 patients without chest radiology finding: 2 cases of PTB had bronchoalveolar fluid positive (PCR technique) for M. tuberculosis; 1 case of EPTB had both sputum and BAL negative for M. tuberculosis; 1 case of intestinal TB was diagnosed on autopsy, patient died after a brief period of intensive care unit admission, no radiologic exam could be performed.
Now in table 1 it is clarified that the number of chest findings concerns 161 patients.
The authors described a 5.7% of isoniazid resistance, I am interested about the method used to determine the isoniazid drug-susceptibility. The methods described by the authors included genotypic methods (Xpert MTB/RIF, and anyPlex) and phenotypic methods. It is strange that the isoniazid, rifampicin, ethambutol, pyrazinamide and aminoglycosides have the same denominators of 122. If the specimen did not underwent DST for the specific agents, it is not appropriate to include this specimen in the denominator.
Answer- Thanks for pointing this out. Twelve of the 16 drug-resistant TB were diagnosed with culture phenotype methods. In the other 4 cases, with cultures negative, 2 cases are diagnosed by Xpert MTB / RIF and in 2 by Anyplex methods. Five of the 7 cases of INH resistance were obtained with phenotypic methods and two with the Aniplex method. We have modified table 1, specifying the results of drug resistance based on the methods used.
There is a small table under the Figure 2(B) describing mixed extra-pulmonary anatomical distribution of 5 patients. The table is not referred by the text.
Answer- we agreed with the Reviewer and apologize for forgetting to refer the table that now it is mentioned in the text (line 138-139). We better specified in the title that the table represents EPTBs with more than one anatomical site.

Reviewer 2 Report
It is a manuscript very interesting, well written, multi-center, and on 5 years period. Well done
I have only some minor suggestions
1: Introduction: well done. Can you explain better which are the High TB-burden countries? And the different incidence beetwen these country and Italian casistic
2. Methods: Well written. No suggestions
3. Results: Your results are very important because underline how this diseases has an increase Temporal trends of admissions. Table 1 and 2 are very clear. Do you have data about educational level and income?
4. Discussion and Conclusions: Well done. If you agree discussion better the role of Social determinants of health on onset Tb and onset of MDR TB, and the comorbidities (especially diabetes).
Author Response
Manuscript ID: ijerph-656990
Type of manuscript: Article
Title: Tuberculosis-related hospitalizations in a low-incidence country: a retrospective analysis in two Italian infectious diseases wards
Reviewer 2
I have only some minor suggestions
1: Introduction: well done. Can you explain better which are the High TB-burden countries? And the different incidence between these country and Italian casuistic
Answer - Thanks for the suggestion. The World Health Organization (WHO) defines high TB-burden countries depending on TB incidence. In 2016 the countries list was updated, now including 30 countries (from 22 previously included), defined as “the top 20 countries in terms of the absolute number of estimated incident cases, plus the additional 10 countries with the most severe burden in terms of incidence rates per capita that do not already appear in the top 20 and that meet a minimum threshold in terms of their absolute numbers of incident cases (10.000 per year for TB, and 1.000 per year for TB/HIV and MDR-TB). The lists were defined using the latest estimates of TB disease burden available in October 2015”.
Low- TB burden countries are defined as having less than 10 cases per 100,000 population per year. In Italy, there were 4,100 new cases estimated in 2017, in a population of almost 6o million people, with an estimated incidence of 7.4 new cases/100.000/year, meeting criteria for low TB incidence country.
To make this clearer, the following sentence was added in the first introduction paragraph “(20 countries with the highest absolute number of TB and 10 countries defined by >10 000 estimated incident TB cases/year)”, lines 34-35.
Methods: Well written. No suggestions
Results: Your results are very important because underline how this diseases has an increase Temporal trends of admissions. Table 1 and 2 are very clear. Do you have data about educational level and income?
Answer – We agree with the reviewer that educational level, income and occupation of the study population would add important and interesting information, however, unfortunately, the requested data were missing on the medical records of the patients enrolled in the study and, given the retrospective nature of the study, it is not possible to recover them now.
Discussion and Conclusions: Well done. If you agree discussion better the role of Social determinants of health on onset Tb and onset of MDR TB, and the comorbidities (especially diabetes) .
Answer – Thanks for the suggestion. A phrase (and a reference) on factors associated with drug resistant TB in European countries is now present in discussion section (lines 201-205)

Round 2
Reviewer 1 Report
Thank you for the revision. I would like to point out several points for this revision.
I suggest to remove the p-values in the Figure 1. These statistics do not help the readers to understand the meaning of change in proportion of extra-pulmonary tuberculosis. I would also suggest to use the stack proportional bar chart to present the percentage of pulmonary, mixed pulmonary and extra-pulmonary, and extra-pulmonary tuberculosis. The sum of the percentages of the three types of tuberculosis should be 100%. The first sentence in the result section described an increase in diagnosis of tuberculosis during 2013-2017. The increase of the diagnosis of tuberculosis was noted in 2013-2016, not in 2017. Furthermore, this finding should refer to Table 1 instead of Figure 1. Most of the information of Table 1 and Table 2 are redundant. The authors can combing these two tables into one, by given three columns (Italian, non-Italian, Overall). I still feel extremely confused about the results of the drug-susceptibility tests. Do you mean, for the 166 patients included, only 12 patients underwent phenotypic drug-susceptibility tests, and only 4 patients underwent Xpert/Anyplex tests? Which drug do you test for phenotypic drug-susceptibility tests? Were isoniazid, rifampicin, ethambutol and streptomycin included in the phenotypic drug susceptibility tests? You should check the agreement of the number of positive results and the percentage of the positive results. It is equally confusing for the percentage of patients underwent 2nd-line anti-TB treatment. For the Table 4, please put the age as categorical variables into the logistic regression model. In the discussion section, line 224-233, the authors described that extra-pulmonary tuberculosis patients were young subjects with few comorbidities. However, refer to Table 3, 43.5% of patients with extra-pulmonary tuberculosis were HIV-positive and 21.7% of them were with hematological malignancies. The authors should consider to rephrase their interpretation for the impact of these comorbidities on the presentation of extra-pulmonary tuberculosis.Author Response
International Journal of Environmental Research and Public Health
Manuscript ID: ijerph-656990R1
Type of manuscript: Article
Title: Tuberculosis-related hospitalizations in a low-incidence country: a retrospective analysis in two Italian infectious diseases wards
Reviewer 1
Thank you for the revision. I would like to point out several points for this revision.
I suggest to remove the p-values in the Figure 1. These statistics do not help the readers to understand the meaning of change in proportion of extra-pulmonary tuberculosis. I would also suggest to use the stack proportional bar chart to present the percentage of pulmonary, mixed pulmonary and extra-pulmonary, and extra-pulmonary tuberculosis. The sum of the percentages of the three types of tuberculosis should be 100%.Answer - We thank the Reviewer for the suggestion. We include the new bar chart with the proportional distribution of cases of pulmonary and extra-pulmonary TB. We would like to highlight that the combined type in the previous figure was only the sum of proportional figures of pulmonary and extra-pulmonary cases, with the exclusion of mixed forms involving the lungs and extra-pulmonary anatomical sites. This new figure 1 is now cited in the lines 138-139.
The first sentence in the result section described an increase in diagnosis of tuberculosis during 2013-2017. The increase of the diagnosis of tuberculosis was noted in 2013-2016, not in 2017. Furthermore, this finding should refer to Table 1 instead of Figure 1. Most of the information of Table 1 and Table 2 are redundant. The authors can combing these two tables into one, by given three columns (Italian, non-Italian, Overall).
Answer - We thanks the Reviewer for the suggestion and the previous Table no. 2 is now Table no. 1. We deleted the previous Table no. 1 as recommended to avoid redundant information. The additional information on not statistically compared variables is now included in the Supplementary materials as supplementary table 1. Supplementary table 1 have been mentioned in the text where appropriate. Furthermore, we edited the text for the year 2017.
I still feel extremely confused about the results of the drug-susceptibility tests. Do you mean, for the 166 patients included, only 12 patients underwent phenotypic drug-susceptibility tests, and only 4 patients underwent Xpert/Anyplex tests? Which drug do you test for phenotypic drug-susceptibility tests? Were isoniazid, rifampicin, ethambutol and streptomycin included in the phenotypic drug susceptibility tests? You should check the agreement of the number of positive results and the percentage of the positive results. It is equally confusing for the percentage of patients underwent 2nd-line anti-TB treatment.
Answer - Thanks for the request of clarification. Data on TB drug-susceptibility were available for 126 patients, of whom 16 had resistant Mycobacterium tuberculosis (MTB) strains. This in now specified in results section (lines 127-128). Twelve of the resistant MTB were identified by phenotypic susceptibility testing (among 98 patients who underwent phenotypic exam) and 4 by molecular methods (among 31 patients with molecular test done). In 3 cases we have both phenotypic and molecular methods drug resistance results.
Phenotypic method evaluate susceptibility to rifampicin, isoniazid, pyrazinamide, ethambutol, and streptomycin; molecular methods evaluate susceptibility to rifampicin, isoniazid, fluoroquinolones and aminoglycosides.
Absolute numbers and percentages of therapies have been corrected in the actual supplementary Table 1. Second-line therapy refers to the treatment of patients who have started with one or more second-line drugs (other than rifampicin, isoniazid, pyrazinamide, ethambutol) in the composition of their tuberculosis treatment. This is now specified in a note at the bottom of the supplementary table 1.
4: For the Table 4, please put the age as categorical variables into the logistic regression model.
Answer - We thank the Reviewer for the suggestion. The categorical age was included in the logistic regression model (table 3).
In the discussion section, line 224-233, the authors described that extra-pulmonary tuberculosis patients were young subjects with few comorbidities. However, refer to Table 3, 43.5% of patients with extra-pulmonary tuberculosis were HIV-positive and 21.7% of them were with hematological malignancies. The authors should consider to rephrase their interpretation for the impact of these comorbidities on the presentation of extra-pulmonary tuberculosis
Answer -We thank the Reviewer for his/her suggestion. We slightly edited the text because the variable immunodepression was detected in 23/61; if we consider 23 immunocompromised cases, 15 were HIV-positives or affected by a hematological malignancy. Then, only one third of the extra-pulmonary TB cases showed an immunodeficiency. Moreover, as specified in the text, the EPTB population is mainly composed of young extra-Europeans men, the majority of African origin (24/61, 39%), without comorbidities, while EPTB cases with comorbidities, namely HIV and hematological diseases, were of Italian origin.
